# AD-H: Autonomous Driving with Hierarchical Agents

## Abstract

Due to the impressive capabilities of multimodal large language models (MLLMs), recent works have focused on employing MLLM-based agents for autonomous driving in large-scale and dynamic environments. However, prevalent approaches often directly use MLLMs to translate high-level instructions into low-level vehicle control signals. This approach deviates from the inherent language generation paradigm of MLLMs and fails to fully harness their emergent capabilities. As a result, the generalizability of these methods is limited by the autonomous driving datasets used during fine-tuning. To tackle this challenge, we propose AD-H, a hierarchical framework that enables two agents (the MLLM planner and the controller) to collaborate. The MLLM planner perceives environmental information and high-level instructions to generate mid-level, fine-grained driving commands, which the controller then executes as actions. This compositional paradigm liberates the MLLM from low-level control signal decoding, thus fully leveraging its high-level perception, reasoning, and planning capabilities. Furthermore, the fine-grained commands provided by the MLLM planner enable the controller to perform actions more effectively. To train AD-H, we build a new autonomous driving dataset with hierarchical action annotations encompassing multiple levels of instructions and driving commands. Comprehensive closed-loop evaluations demonstrate several key advantages of our proposed AD-H system. First, AD-H can notably outperform state-of-the-art methods in achieving exceptional driving performance, even exhibiting self-correction capabilities during vehicle operation, a scenario not encountered in the training dataset. Second, AD-H demonstrates superior generalization under long-horizon instructions and novel environmental conditions, significantly surpassing current state-of-the-art methods.

## 1 Introduction

Autonomous driving systems represent a major advancement in contemporary transportation, which requires vehicles to automatically operate in *large-scale* and *dynamic* environments. With the rapid advancement of Multimodal Large Language Models (MLLMs) (Liu et al., 2024; Dai et al., 2024; Li et al., 2023a; Yin et al., 2024; Zhang et al., 2023c; Zhu et al., 2023) and MLLM-based agents (Driess et al., 2023; Brohan et al., 2022; 2023; Belkhale et al., 2024; Wang et al., 2023a;g; Lifshitz et al., 2024; Qin et al., 2023b; Zhou et al., 2024a), recent attempts (Sima et al., 2023; Wang et al., 2023b; Shao et al., 2023; Chen et al., 2023b; Liu et al., 2023a; Sha et al., 2023; Wen et al., 2023a; Tian et al., 2024) have been made to explore MLLMs as the central agent of autonomous driving systems for better perception, reasoning, and interactions, which have achieved remarkable progress. A predominant paradigm adopted by these methods is to translate high-level contextual instructions into low-level control signals using MLLMs. As MLLMs are pre-trained to generate natural languages, their ability to decode low-level control signals is highly reliant on the autonomous driving datasets used during fine-tuning, causing significant overfitting to specific scenarios and instructions. As an example, Figure 1 (a) depicts an oversteering scenario that is absent in the training dataset. Most existing methods struggle to adapt to this case and often maintain straight motion even after excessive turning, leading to dangerous situations. These limitations motivate us to delve into an intriguing and pivotal question: *Is it possible to develop an autonomous driving system that can fully unleash the emergent capabilities of pre-trained MLLM for more intelligent reasoning and stronger scalability towards unseen scenarios and instructions?*

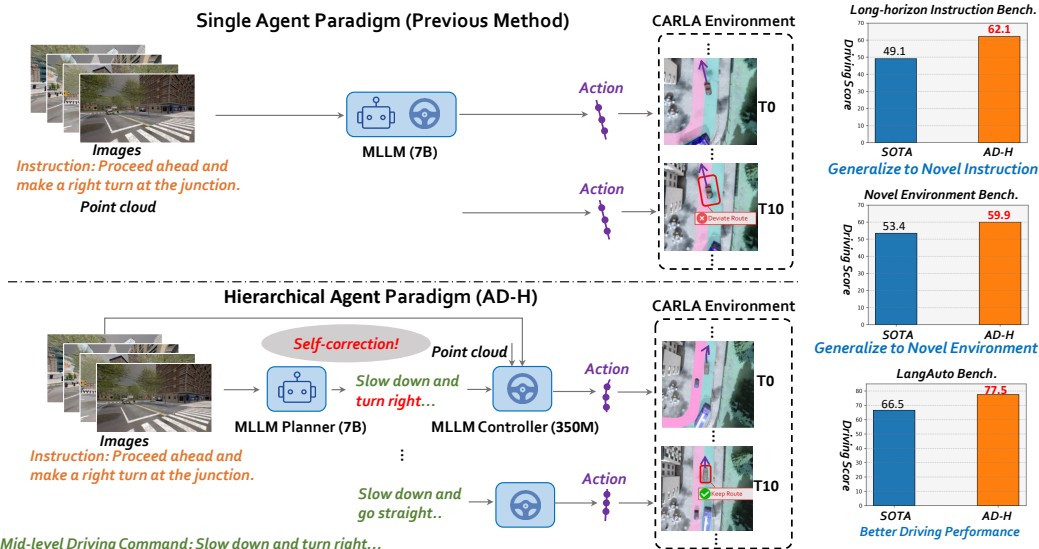

Figure 1: This figure compares the previous single agent paradigm with our hierarchical multi-agent paradigm (AD-H), emphasizing compositional task handling for autonomous driving. The single-agent method directly converts high-level instructions into actions, while AD-H decomposes tasks into mid-level commands via a planner-controller structure, enabling better compositional reasoning. The graphs on the right highlight AD-H's superior performance in generalizing to novel instructions, unseen environments, and overall driving capabilities compared to the SOTA.

To answer the above question, we explore the the concept of compositional paradigm (Du & Kaelbling, 2024) and hierarchical policy (Belkhale et al., 2024; Chen et al., 2024). Instead of predicting the final control signals directly with a single MLLM, we propose a collaborative approach using two models. The workflow transitions from high-level instructions to mid-level driving commands, and finally to low-level actions. On the one hand, compared to high-level contextual instructions, mid-level commands offer a finer granularity and lie closer to low-level control signals, permitting more precise reflection on real-time environmental feedback. On the other hand, different from low-level control signals, mid-level commands are natural language-driven and are therefore better aligned with the pre-training target of MLLMs to leverage their world knowledge. In addition, breaking down high-level instructions into mid-level commands further enables more flexible human interaction and effective shared policy structure learning across similar tasks (Belkhale et al., 2024), giving rise to stronger generalization abilities to novel instruction and scenarios.

In light of the above motivation, we design a Hierarchical Multi-Agent System for Autonomous Driving (AD-H), which comprises two agents: a MLLM-based planner and a lightweight controller. As shown in Figure 1 (b), the planner aims to perform planning and decision-making based on the input contextual high-level instruction and predicts a mid-level command at each decision frame. The mid-level command is then decoded into the low-level control signals by the controller given the current visual input and the contextual instruction. The high-level planner and low-level controller together form a hierarchical policy system, which effectively frees the MLLM from low-level decoding and unlocks its potential for high-level perception, reasoning, and planning. The last issue remaining is the lack of annotated data for training the hierarchical systems, as existing autonomous datasets do not contain mid-level commands. To this end, derived from LMDrive dataset (Shao et al., 2023), we further build a new training dataset including 1,753K frames with hierarchical annotations encompassing multi-level instructions and commands.

Through intensive evaluations under the closed-loop environment, we show that our AD-H enjoys the following two advantages. *First, AD-H can better generalize to novel scenarios.* Since the high-level reasoning and low-level execution are decoupled in our hierarchical multi-agent system, the planner solely focusing on high-level reasoning can more effectively leverage the emergent capability of pre-trained MLLMs, yielding stronger generalization power and reasoning ability under unseen driving scenarios and even challenging corner cases. For example, in cases of oversteering, the planner issues corrective instructions to guide the vehicle back on the right track (Figure 1 (b)). In contrast,

previous methods tend to severely overfit to control signal patterns within the training set, resulting in a tendency to persistently move straight (Figure 1 (a)). As a result, AD-H achieves a notable improvement in driving performance compared to state-of-the-art methods. *Second, AD-H can better generalize to novel long-horizon instructions.* Our long-horizon experiments reveal that AD-H can comprehensively understand novel long-horizon instructions, perform effective planning, and generate precise driving commands at appropriate decision frames. This has led to a significant improvement in performance for long-horizon tasks. In contrast, existing methods show poor generalization to long-horizon instructions, often resulting in erroneous routes.

The contribution of this paper can be summarized as follows:

- We propose AD-H, a hierarchical multi-agent system for autonomous driving, which can significantly unleash the power of MLLMs to achieve higher control precision and generalization.
- We construct an autonomous driving dataset with 1,753k multi-level driving command annotations, which can effectively facilitate hierarchical policy learning.
- We perform intensive experiments and demonstrate that our approach can considerably outperform state-of-the-art methods and exhibits stronger generalization to novel scenarios and long-horizon instructions.

## 2 RELATED WORKS

### 2.1 END-END METHODS IN AUTONOMOUS DRIVING

In autonomous driving, precise perception (Li et al., 2022d; Yang et al., 2023; Liu et al., 2023b; Philion & Fidler, 2020; Liang et al., 2022; Qin et al., 2023a; Li et al., 2022a; Jiao et al., 2023; Yoo et al., 2020; Li et al., 2022b; Bai et al., 2022; Chen et al., 2022; Huang et al., 2021; Li et al., 2022c; Park et al., 2022; Li et al., 2023d; Zhou et al., 2023a; Wang et al., 2023f;e; 2024a; Zhang et al., 2023b; Ge et al., 2023; Li et al., 2023c) and planning are critical. To tackle the prevalent issue of long-tail distribution in autonomous driving scenarios, several generative network-based World Models have been developed (Wang et al., 2023c; Jia et al., 2023; Zhao et al., 2024; Wen et al., 2023b). These networks can generate a vast array of realistic urban street scenes. However, in order to control the vehicle, a separate planning model needs to be designed to utilize the perception results. To solve this problem, many end-to-end autonomous driving models have been proposed, including reinforcement learning based (Prakash et al., 2021; Wu et al., 2022; Chitta et al., 2022; Codevilla et al., 2019; Cui et al., 2022) and imitation learning based methods (Xiao et al., 2023; Hanselmann et al., 2022). Besides these, UniAD (Hu et al., 2023a) addresses the problem of end-to-end autonomous driving by utilizing multiple modules in BEV space.

Since the emergence of multimodal large models, the field of autonomous driving has been continuously exploring the possibility of using such large models in an end-to-end manner to solve this problem. LLM-Driver (Chen et al., 2023a) uses Vector-former to characterize the perception of the environment by autonomous driving in vector space. Drivegpt4 (Xu et al., 2023) proposes a novel two-stage training multimodal autonomous driving paradigm, which directly regresses control signals and text responses through multi-frame image input and text instructions. DOLPHINS (Ma et al., 2023) innovatively introduces in-context learning into the autonomous driving framework, which can better mimic human higher-order control abilities. Unlike the methods mentioned above that are trained and tested on static datasets, LMDrive (Shao et al., 2023) first conducts closed-loop autonomous driving training and testing on the Carla simulator, demonstrating strong closed-loop control capabilities and scene generalization. As well as several other notable contributions in this area (Li et al., 2024; Zhou et al., 2023b; Ding et al., 2024; Wang et al., 2023d; Ye et al., 2024; Peng et al., 2024; Paul et al., 2024; Wang et al., 2024b). Besides, there have been some exploratory endeavors to leverage agent-based approaches in the domain of autonomous vehicular navigation (Yang et al., 2024; Mao et al., 2023).

### 2.2 MULTIMODAL LARGE LANGUAGE MODELS

Multimodal Large Language Models (MLLMs) have garnered considerable attention for their remarkable abilities in multimodal perception. Several studies (Liu et al., 2024; Dai et al., 2024; Zhang

et al., 2023c; Zhu et al., 2023; Lai et al., 2023; Peng et al., 2023) focus on integrating visual content into language models, specifically designed to comprehend and reason about images. Among these, LLaVA (Liu et al., 2024) employs a two-stage instruction-tuning pipeline for comprehensive visual and language understanding. InstructBLIP (Dai et al., 2024) combines the language model with an instruction-aware Q-Former to extract visual content highly pertinent to the provided instruction. Additionally, research (Deshmukh et al., 2023; Li et al., 2023b; Zhang et al., 2023a; Guo et al., 2023; Hong et al., 2023) is expanding MLLMs to include audio, video, and point clouds, enhancing their ability to handle complex multimodal tasks. This integration allows MLLMs to process spatial, auditory, and visual data simultaneously, significantly improving performance in applications like autonomous navigation and multimedia analysis.

### 2.3 LLMs in Task Planning

In various fields, LLMs have demonstrated their potential in task decomposition for advanced planning. LLMs can incorporate additional visual modules, such as caption descriptions, to perceive environments and influence planning outcomes. SayCan (Ahn et al., 2022b) integrates LLMs with robotic capabilities, allowing robots to follow complex, long-term natural language instructions. Here, the LLM provides a high-level understanding of the instructions and identifies skills that can offer corresponding low-level controls. To avoid error accumulation due to model stacking, recent research has explored using MLLMs for planning. ViLa (Hu et al., 2023b) leverages the world knowledge inherent in MLLMs, including spatial layouts and object attributes, to make more rational task planning for manipulative tasks. RT-H (Belkhale et al., 2024) improves task execution accuracy and learning efficiency by decomposing complex tasks into simple language instructions that are then converted into robotic actions. Nevertheless, it has mainly been investigated under small-scale and static scenarios. It is unknown whether this philosophy can also generalize to large-scale and dynamic autonomous driving environments. More importantly, it lacks suitable training datasets for learning such systems. Our work has filled the above gaps.

## 3 Method

In this section, we will first delineate the technical details of our proposed AD-H autonomous driving system, and then present the new dataset for training hierarchical multi-agent systems.

### 3.1 Method Overview

The AD-H system consists of two MLLM-based agents, namely a planner and a controller, as illustrated in Figure 2 (a). At each decision frame, the planner consumes the current visual input and a high-level contextual instruction (*e.g.*, "turn left at the next intersection"), performs reasoning & planning, and makes a decision for the current frame by predicting a mid-level driving command (*e.g.*, "slow down to ensure safety"). The controller then receives the predicted command and converts it into future waypoints to control the vehicle. The planner and controller, together with the input high-level instruction, the predicted mid-level commands, and low-level waypoints form a hierarchical structure of action policy for autonomous driving. The overall pipeline can be mathematically expressed as

$$\mathbf{y}_t = g(f(\mathbf{x}_t, \mathbf{i}), \mathbf{x}_t, \mathbf{i}), \tag{1}$$

where $\mathbf{i}$ denotes the contextual driving instruction, $\mathbf{x}_t$ and $\mathbf{y}_t$ denotes the visual input and the predicted control signals (*i.e.*, waypoints) for the $t$-th frame, respectively, and $f$ and $g$ represent the high-level planner and low-level controller, respectively.

### 3.2 High-level Planner

In the AD-H system, the planner focuses solely on high-level decision-making without getting involved in the generation of low-level control signals and therefore becomes more specialized. To do so, the planner needs to perform not only visual perception to understand the surrounding environment as well as its ego status but also effective reasoning and planning to break down the contextual instruction into mid-level driving commands. To this end, we adopt a MLLM as the high-level planner to leverage their strong emergent capabilities (We mainly explore LLaVA-7B (Liu et al., 2024) and Mipha-3B (Zhu et al., 2024) in our experiments). Figure 2 (a) illustrates an overview

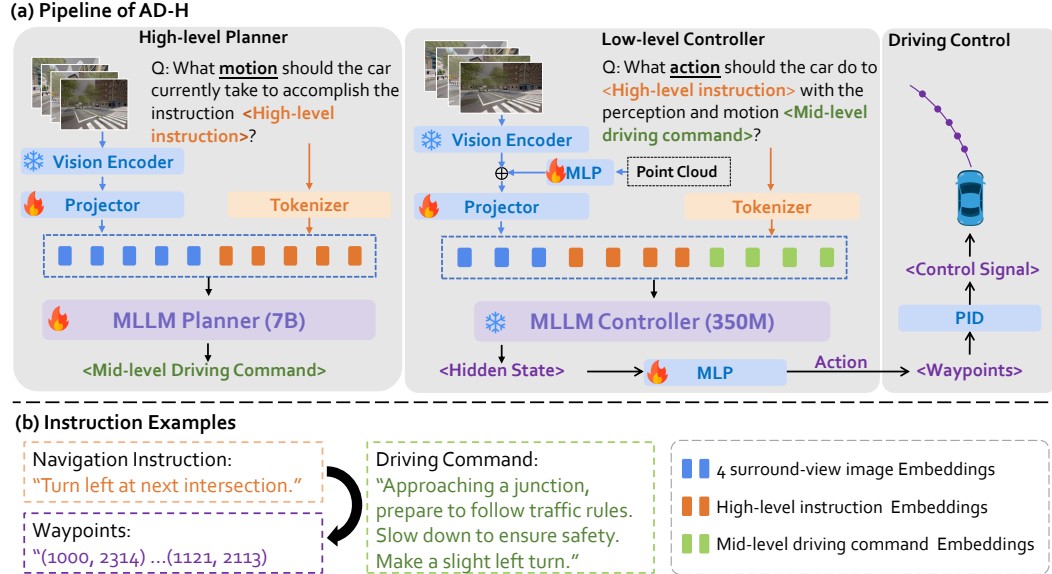

Figure 2: (a) Pipeline of AD-H. The planner breaks down a high-level instruction into mid-level driving commands and the controller decodes low-level waypoints from the mid-level commands. (b) Examples of a high-level instruction, a mid-level command, and low-level waypoints.

of the MLLM-based planner. At each decision frame, 4 surround-view images are concatenated and fed into a pre-trained vision encoder (Radford et al., 2021). The encoded visual features are further transformed into the textual token space through a projector. Finally, the visual feature together with the tokenized high-level instruction are sequentially fed into the MLLM to predict the mid-level command in an auto-regressive manner.

Through internet-scale pre-training and massive instruction tuning, MLLMs have acquired powerful reasoning ability, along with a wealth of world knowledge, which allows MLLMs to generalize better across various tasks and application scenarios. We then proceed with downstream fine-tuning on our collected autonomous driving dataset (Section 4) to teach MLLMs how to generate precise mid-level commands through the next token prediction given the contextual information. Since the driving commands are also natural languages, this downstream task is essentially consistent with the pre-training objectives of MLLMs. As such, the emergent capabilities of the pre-trained MLLMs can be fully unleashed. Our experiments show that the MLLM-based planner can better generalize to novel driving scenarios, long-horizon instructions, as well as unseen environments, and even exhibits self-correction abilities.

## 3.3 LIGHTWEIGHT CONTROLLER

The role of the controller is to translate the intermediate driving commands generated by the planner into executable control signals, which is much easier than directly predicting the control signals from the high-level instructions. Therefore, instead of using the 7B LLaMA model (Liu et al., 2024) as in LMDrive (Shao et al., 2023), we adopt the more lightweight OPT-350M (Zhang et al., 2022) for this purpose. Since OPT-350M is a pure language model, we empower it with visual perception ability by adding an additional vision encoder (He et al., 2016) and a Q-Former (Li et al., 2023a). As shown in Figure 2, the pipeline of the controller is similar to that of the planner. The input images are also encoded by the vision encoder and then concatenated with the point cloud features. The concatenated features are projected into the space of textual features through the pre-trained Q-Former. OPT-350M then receives the visual embeddings as well as the textual tokens of the high-level instructions and mid-level commands. The hidden state of its output layer serves as the action embedding and is finally decoded into 5 future waypoints through 2-layer MLP. These waypoints can be input into downstream control algorithms (e.g., PID) to produce numerical information for vehicle control, such as speed, throttle, and steering angle. The above pipeline for the controller can be mathematically

expressed as

$$\mathbf{h}_t = g_l(\mathbf{x}_t, \mathbf{i}, \mathbf{c}_t), \tag{2}$$

$$\mathbf{y}_t = g_w(\mathbf{h}_t), \tag{3}$$

where $\mathbf{c}_t$ represents the mid-level command generated by the planner, $g_l$ and $g_w$ denote the OPT-350M model and the MLP for waypoint regression, respectively, and $\mathbf{h}_t$ indicates the hidden state output of OPT-350M. During training, we feed the ground-truth mid-level command into the controller and minimize the $L_1$ loss between the predicted and ground-truth waypoints.

## 4 TRAINING DATASET CONSTRUCTION

The mid-level driving commands play a pivotal role in training a proficient planner and controller. Recent works (Sima et al., 2023; Shao et al., 2023; Zhou et al., 2024b) have collected a significant amount of image and instruction data in real-world scenarios and closed-loop simulators like CARLA (Dosovitskiy et al., 2017). However, these datasets lack consideration for mid-level driving commands, rendering training impractical. To address this issue, we create a novel action hierarchy dataset LMDrive-H derived from LMDrive dataset (Shao et al., 2023). Our dataset comprises annotations across three distinct hierarchical levels: high-level instructions, mid-level driving commands, and low-level vehicle control signals. Initially, we extract about 160k video-instruction pairs from LMDrive dataset (Shao et al., 2023), alongside low-level vehicle control signals for each frame. Subsequently, leveraging the detailed measurement record provided by CARLA (Dosovitskiy et al., 2017) for each frame, including throttle, speed, steering angle, etc., we employ a rule-based methodology (See Supplementary Materials) to retrospectively deduce the mid-level driving commands for each frame.

Specifically, we first develop a comprehensive set of driving commands. Autonomous driving, unlike robotic grasping scenes, takes place in a dynamic and complex environment, necessitating a more fine-grained command construction than merely selecting actions like acceleration, braking, or turning left. Our fine-grained driving command encompasses both perceptual information and motion details, including crucial data about pedestrians, vehicles, and road signs. This approach aligns with the structure of LLMs and reflects the chain of thought ideology (Wei et al., 2022; Sima et al., 2023). After resampling, we obtain 1,753K hierarchal annotations. More details about our dataset are presented in Supplementary Materials A.2.

## 5 EXPERIMENTS

### 5.1 EXPERIMENTAL SETTINGS

#### 5.1.1 IMPLEMENTATION DETAILS.

Our main experiments are achieved by using the pre-trained LLaVA-7B-V1.5 (Liu et al., 2024) with ViT (Dosovitskiy et al., 2020) vision encoder and the OPT-350M (Zhang et al., 2022) with a ResNet50 vision encoder as the high-level planner and low-level controller, respectively. We also explore other MLLM architectures (Liu et al., 2024; Zhu et al., 2024) in Section ??. Unless otherwise stated, the AD-H is fine-tuned on our LMDrive-H dataset with only vision encoders fixed. For the high-level planner, the initial learning rate is set to 2e-5, and a few steps of warm-up are incorporated into the training process. For the low-level controller, the learning rate is set to 1e-5. Training is conducted using the Adam optimizer with a batch size of 32 on 4 NVIDIA A800 GPUs. Please see Supplementary Materials A.1 for more implementation details.

#### 5.1.2 EVALUATION BENCHMARKS AND METRICS

We conduct standard closed-loop evaluations using CARLA simulator (Dosovitskiy et al., 2017) on the LangAuto Benchmark (Shao et al., 2023). On top of LangAuto, we further build two additional benchmarks termed LangAuto-Long-Horizon and LangAuto-Novel-Environment, which contain long-horizon instructions and novel environments, respectively. We present their details as follows.

**LangAuto Benchmark.**    The LangAuto benchmark encompasses a variety of test routes spanning eight towns, diverse weather conditions, and misleading interference. Throughout the testing procedure, algorithms navigate vehicles within the environment, utilizing solely language commands and visual input. The LangAuto benchmark is further divided into three sub-tracks: LangAuto, with routes longer than 500 meters; LangAuto-Short, with routes between 150 and 500 meters; and LangAuto-Tiny, with routes shorter than 150 meters. We follow the prior method (Shao et al., 2023) and perform evaluations separately on these three sub-tracks.

**LangAuto-Long-Horizon Benchmark.**    Planning and decision-making over long-time horizons is a central concern in embodied AI (Pirk et al., 2020; Huang et al., 2022a; Zeng et al., 2022; Ahn et al., 2022a; Huang et al., 2022b), which typically necessitate a series of sub-instructions to fulfill a primary goal. To ascertain the effectiveness of AD-H in such scenarios, we have built LangAuto-Long-Horizon based on the LangAuto-Tiny Benchmark by combining multiple instructions to form long-horizon instructions. For instance, the instruction series "Alright, you can start driving", "Keep on rolling straight till you get to the next junction," and "Continue in a straight line along your current path" are condensed into a streamlined directive: "Go straight ahead, turn left at the end of the road, then continue straight." Additionally, given that neither our approach nor the baseline model incorporates historical frame information, we include environmental cues in long-horizon instructions to avoid ambiguity (such as uncertainty about whether to turn at a particular intersection). For instance, "Go straight until you see a turning point with palm trees ahead, then turn right and follow the road." Details of long-horizon instructions are presented in Supplementary Materials A.4.2. Since all the long-horizon instructions are absent from the LMDrive-H training set, the LangAuto-Long-Horizon benchmark can also verify the generalization ability of autonomous driving systems to novel instructions.

**LangAuto-Novel-Environment Benchmark.**    To evaluate the generalization ability of autonomous driving systems under new environments, we have built LangAuto-Novel-Environment based on the LangAuto-Tiny Benchmark by only retaining driving routes from 7 out of 8 Towns (Town02-07, 10). To ensure non-overlap between training and testing, we have further removed training data belonging to the above 7 Towns from the LMDrive-H training set.

**Evaluation Metrics.**    We employ three widely used evaluation metrics from the CARLA Leader-Board (Dosovitskiy et al., 2017), including route completion (RC), infraction score (IS), and driving score (DS). Among them, RC measures the percentage of the planned route that is successfully completed, with a specific focus on the distance covered along designated segments. Any significant deviation from the intended route leads to the episode being marked as a failure. The IS metric keeps track of violations such as collisions or traffic infractions, which decrease the score with each offense. The DS metric combines both the RC and IS scores to provide a comprehensive assessment of progress and safety, serving as the primary evaluation metric.

## 5.2 RESULTS AND ANALYSIS

In this section, we mainly investigate the performance of the autonomous driving models from four perspectives: (1) standard evaluation in a closed-loop manner, (2) generalization to novel long-horizon instructions, (3) generalization to novel environments, and (4) performance achieved by using different MLLMs as planners. As the LangAuto is a new benchmark, only the result of LMDrive (Shao et al., 2023) is available. Therefore, we adopt LMDrive as our main competitor. It should be noted that LMDrive is one of the pioneering works in language-guided closed-loop driving and can serve as a strong baseline of our method without using hierarchical agents.

### 5.2.1 CLOSED-LOOP DRIVING PERFORMANCE

Table 1 reports the quantitative comparisons on the LangAuto benchmark. It shows that our AD-H significantly outperforms LMDrive for the three different sub-tracks, especially in terms of the main score DS, indicating that the mid-level driving commands generated by our planner enable the controller to act more accurately within large-scale and complex environments. Moreover, we find that even the smaller models (Mipha-3B and OPT-350M) perform significantly better than the 7B model, which further validates the effectiveness of the AD-H hierarchical paradigm. Through

extensive analysis, we further observe that AD-H exhibits frequent self-correction behaviors. As shown in Figure 3, LMDrive fails to recognize the road conditions after a left turn, causing the vehicle to continue maintaining the steering wheel straight as the previously received high-level navigation instruction. Consequently, the vehicle crosses the lane boundary and deviates from the correct path. In contrast, our AD-H utilizes its planner to dynamically generate mid-level commands, enabling the controller to adjust its posture accordingly, which effectively reduces the risk of traffic jams and accidents. More visualizations are provided in the Supplementary Materials A.4.2.

Table 1: Comparison on LangAuto benchmark.

| Method | LangAuto | | | LangAuto-Short | | | LangAuto-Tiny | | |
|---|---|---|---|---|---|---|---|---|---|
| | DS(↑) | RC(↑) | IS(↑) | DS(↑) | RC(↑) | IS(↑) | DS(↑) | RC(↑) | IS(↑) |
| LMDrive (LLaVA-7B) (Shao et al., 2023) | 36.2 | 46.5 | 0.81 | 50.6 | 60.0 | 0.84 | 66.5 | 77.9 | 0.85 |
| AD-H (Mipha-3B + OPT-350M) | 41.1 | 48.5 | **0.86** | 54.3 | 61.8 | **0.86** | 68.0 | 74.4 | 0.87 |
| AD-H (LLaVA-7B + OPT-350M) | **44.0** | **53.2** | 0.83 | **56.1** | **68.0** | 0.78 | **77.5** | **85.1** | **0.91** |

Figure 3: Results of self-correction scenario. (a) High-level instruction; (b) Visualization results of LMDrive; (c) Visualization results of AD-H; (d) Mid-level driving commands predicted by the planner of AD-H. The visual results show that LMDrive maintains a straight trajectory after oversteering, deviating from the intended path. However, AD-H is able to issue precise commands to guide the vehicle back on track.

### 5.2.2 GENERALIZATION TO LONG-HORIZON INSTRUCTION

Table 2 presents the results on the LangAuto-Long-Horizon benchmark, where the high-level navigation instructions are long-horizon and are provided only at the beginning of the driving task. Considering that both LMDrive and AD-H are trained under short-horizon instructions during the driving process, these unseen long-horizon instruction settings pose a significant challenge in terms of their generalization ability. Nevertheless, our AD-H still delivers strong performance, surpassing the LMDrive method by a considerable margin. As illustrated in Figure 4, LMDrive, which directly predicts control signals, struggles to properly understand the global instructions and road conditions provided in the long-horizon instruction. Consequently, it continues straight instead of making a right turn when necessary. In comparison, the planner of our AD-H continuously analyzes the instructions and the visual environment during the driving process, providing accurate and fine-grained commands to the controller based on the current driving conditions. These results indicate the promising generalization capability of AD-H for unseen navigation instructions.

Table 2: Comparison on LangAuto-Long-Horizon benchmark.

| Method | DS(↑) | IS(↑) | RC(↑) |
|---|---|---|---|
| LMdrive (Shao et al., 2023) | 49.1 | 0.871 | 56.4 |
| AD-H | **62.1** | **0.875** | **68.3** |

Table 3: Comparison on LangAuto-Novel-Environment benchmark.

| Method | DS(↑) | IS(↑) | RC(↑) |
|---|---|---|---|
| LMdrive (Shao et al., 2023) | 53.4 | 0.827 | 64.3 |
| AD-H | **59.9** | **0.875** | **67.8** |

### 5.2.3 GENERALIZATION TO NOVEL ENVIRONMENTS

Table 3 compares AD-H and LMDrive on the LangAuto-Novel-Environment benchmark to assess their zero-shot adaptation capabilities to the unseen environment. Our AD-H consistently outperforms LMDrive across all the metrics, which verifies its strong generalization ability to novel environments.

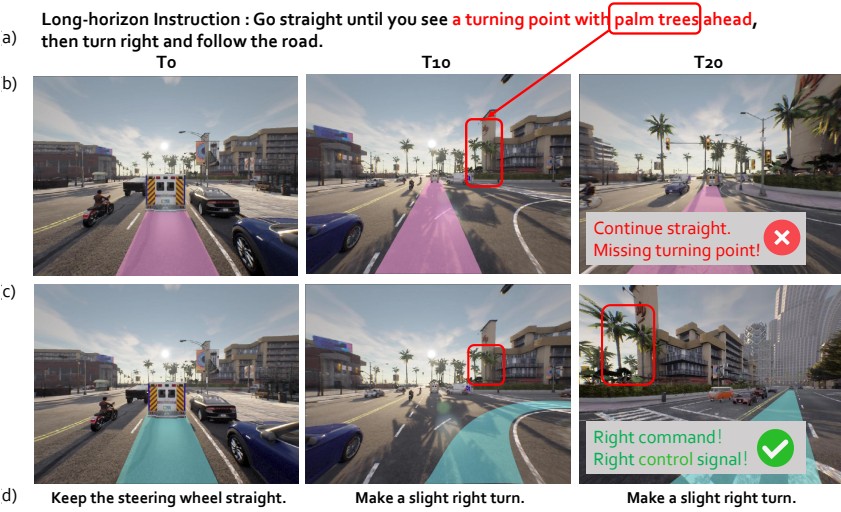

Figure 4: Results with long-horizon instructions (a). (b) LMDrive persists in following the initial instructions, continuing forward; (c) AD-H can adeptly assess environmental cues to determine the appropriate timing for turning; (d) Mid-level commands produced by AD-H.

## 6 ABLATION STUDY

### 6.1 ABLATION ON TRAINING DATASETS

Since our method resampled the LM-Drive dataset, to ensure that the performance improvement is not due to changes in the dataset, we retrained LMDrive using our own dataset. The comparison between the retrained LMDrive model and our method are shown in Table 5. The results demonstrate that resampling the dataset does not significantly enhance performance. Therefore, the improved performance of AD-H is not due to data resampling.

| Method | DS(↑) | IS(↑) | RC(↑) |
|---|---|---|---|
| LMDrive + LMDrive-Dataset | 66.5 | 0.85 | 77.9 |
| LMDrive + ADH-Dataset | 60.7 | 0.91 | 65.7 |
| AD-H + ADH-Dataset | **77.5** | **0.91** | **85.1** |

Figure 5: Comparison between LMDrive and AD-H on different datasets in LangAuto-Tiny Benchmark.

## 6.2 ABLATION ON DIFFERENT CONTROLLERS

We also test controller of different sizes, and the results are shown in Table 6. The following results show that the OPT-350M is comparable with LLaVA-7B, which may be attributed to the fact that the mid-level commands are already very accurate and fine-grained, reducing the burden on the control signal decoding. However, the performance of OPT-125M is unsatisfactory. This unexpected phenomenon warrants further analysis, which we will conduct in future work.

| Method | DS(↑) | IS(↑) | RC(↑) |
|---|---|---|---|
| llava-7b | 74.6 | 0.80 | **90.5** |
| opt-350m | **77.5** | **0.91** | 85.1 |
| opt-125m | 33.9 | 0.90 | 35.9 |

Figure 6: Comparison between different AD-H controllers

## 7 CONCLUSION AND LIMITATION

**Conclusion**    In conclusion, our proposed hierarchical multi-agent driving system, AD-H, bridges high-level instructions and low-level control signals with mid-level language-driven commands. By liberating the multimodal large language models from the burden of decoding low-level control signals, AD-H fully leverages their emergent capabilities in high-level perception, reasoning, and planning. This hierarchical design not only enhances the efficiency and reliability of autonomous driving systems but also enables them to achieve remarkable driving performance even in scenarios not encountered during training. Through comprehensive evaluation, AD-H outperforms the state-of-the-art method, demonstrating remarkable driving performance and adaptability to novel scenarios and instructions. The proposed AD-H harnesses the emergent powers of multimodal large language models, enhancing the efficiency and reliability of autonomous driving systems.

**Limitation**    Given that AD-H operates as a hierarchical agent system, both its size and computational needs are significant. Achieving a lighter version for deployment on actual vehicles will require notable advancements. Moreover, since our experimental data mainly comes from simulations, it's crucial to gather more real-world data to improve domain transfer effectively. Additionally, because virtual scenarios offer limited data diversity, it's urgent to have richer datasets. These datasets are essential for refining instruction tuning and enhancing the capabilities of MLLMs.

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

# A SUPPLEMENTAL MATERIAL

## A.1 IMPLEMENTATION DETAILS

### A.1.1 HIGH-LEVEL PLANNER

**Visual Input.** The motion planner receives visual input from four directional cameras, each capturing an RGB image. To maintain consistency with the pre-trained VLM, we concatenate these four images vertically and feed them into the visual encoder together. This approach offers two advantages: firstly, it aligns with the input format of the pre-trained VLM, preventing confusion that might arise from separate inputs; secondly, it reduces computational complexity by minimizing the token count. Preliminary experiments indicate that combining the four images adequately meets the requirements for input.

**Textual Input.** The planner in AD-H is pivotal as it breaks down high-level navigation instructions into mid-level driving commands. In detail, the textual input of the planner is "What motion should the car currently take to accomplish the instruction <High-level Instruction>?".

**Training.** In our experiments, we employ two scales of MLLMs: LLaVA-7B-V1.5 Liu et al. (2024) and Mipha-3B Zhu et al. (2024). We fine-tune their pre-trained versions on the LMDrive-H dataset for one epoch using $4 \times$ A800s, with the visual encoder kept frozen. During the independent training of the planner, we assess its performance by measuring accuracy on the validation set, as the AD-H system only supports combined testing. The batch size is set to 32, and 3% of the total steps are allocated for warm-up. We utilize the Adam optimizer with an initial learning rate of 2e-5.

### A.1.2 LOW-LEVEL CONTROLLER

**Model.** Similar to the planner, the controller uses the ResNet50 He et al. (2016) model to extract features from images captured from four different angles. textual input of the planner is "What action should the car do to <High-level Instruction> with the perception and motion <Mid-level Driving Command>?". These features are then projected into the controller's input space for the LLM by an adapter made up of MLP, Which are concatenat with motion embeddings, which are processed from driving commands provided by the high-level planner through a tokenizer.We ultimately chose OPT-350m for its optimal balance of performance and speed. The final layer's hidden features from this model are fed into an MLP-based waypoints predictor, which generates the vehicle's position for the upcoming five-time steps. These position details are then translated into direct control signals like steering and throttle through a PID algorithm to interact with the vehicle.

**Training.** Specifically, our experiments are conducted on four A800 GPUs with a batch size of 32. The visual encoder, ResNet50, underwent the same pre-training as used in the LMDrive Shao et al. (2023) . As with the controller, we set the learning rate at 1e-5, with a weight decay of 0.06. Since the controller directly generates waypoints. We train controller with L1 loss and use it as evaluation metrics.

## A.2 DATASET DETAILS

### A.2.1 OVERVIEW

The AD-H Dataset is an innovative action hierarchy dataset specifically designed for autonomous driving. It focuses on mid-level driving commands, making training more practical. Specifically, the AD-H Dataset includes 1.7 million entries, each containing RGB images from four directions, high-level instructions, mid-level driving commands, and low-level vehicle control signals. The process of dataset generation is illustrated in Figure 7.

### A.2.2 DRIVING COMMAND

In our study, we analyze 26 distinct types of driving sub-commands within the AD-H dataset. These sub-commands cover nearly all the key perceptual objects in various driving scenarios and encompass all necessary driving actions. By combining these sub-commands, we generate over 160 different

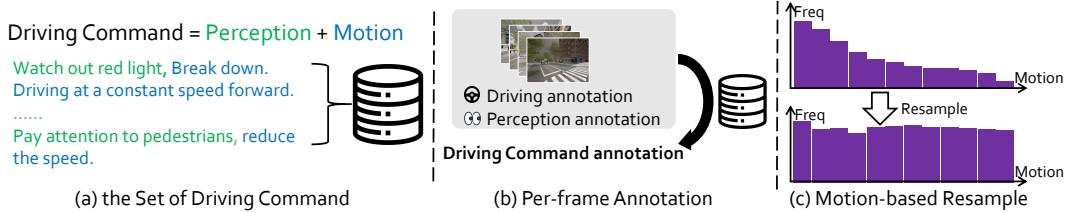

Figure 7: Dataset Generation Pipeline.

variations of driving commands. The complete list of driving sub-commands is provided in Table 9. For instance, when encountering a red light, the appropriate driving command would be, "There is a red light ahead. Apply brakes safely."

### A.2.3  ANNOTATION

The mid-level driving command is determined by the detailed driving data provided by CARLA for each frame, which includes information such as throttle, speed, and steering angle. We employed a rule-based methodology to retrospectively deduce the mid-level driving commands for each frame. For example, when encountering a red light, the appropriate driving command would be, "There is a red light ahead. Apply brakes safely". The data from CARLA provides information about the red light in the scene and whether the vehicle is braking. Details are presented in Table 9.

### A.2.4  RESAMPLING

Initial annotation reveals a significant long-tail distribution issue within the dataset: some motion instructions occur hundreds of times more frequently than others. Common driving scenarios, such as maintaining a steady speed or stopping, predominate, while actions like turning and decelerating are relatively rare. This imbalance can severely impact the model's performance. To address this, we resample the dataset based on the frequency of motion instructions, reducing its size from 3 million to 1.7 million entries, thereby enhancing the dataset's quality.

### A.3  LANGAUTO-LONG-HORIZON BENCHMARK

We present the long-horizon instructions in Table 8.

### A.4  MORE RESULTS

### A.4.1  VISUALIZATION OF AN EXAMPLE

We present a complete example of AD-H, from high-level instructions and sensor input to mid-level driving commands and finally to waypoints, as shown in Table 4.

### A.4.2  MORE VISUALIZATION

We present more visualization results in Figure 4, Figure 5, Figure 6 and Figure 7.

### A.5  SOCIETAL IMPACTS

The proposed approach of utilizing mid-level language-driven commands in autonomous driving systems presents several potential positive societal impacts. By bridging the gap between high-level instructions and low-level control signals, AD-H could lead to safer and more efficient autonomous driving in diverse and dynamic environments. This could ultimately reduce traffic accidents and fatalities, alleviate congestion, and improve accessibility for individuals with mobility limitations. Moreover, the enhanced generalizability of AD-H may foster wider adoption of autonomous vehicles, potentially leading to reduced greenhouse gas emissions and enhanced urban planning.

However, there are also potential negative societal impacts to consider. Dependence on advanced autonomous driving systems like AD-H may exacerbate existing societal issues such as job displace-

Table 4: An example of how our AD-H predicts future waypoints. Our planner provided accurate motion instructions, and the controller accurately execute navigation and motion instructions.

| **Challenging examples of novel and complex environments.** | |
| --- | --- |
| Sensor Input | 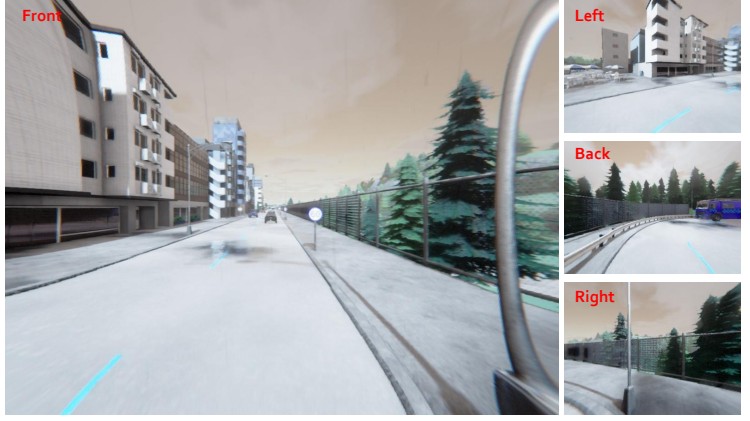 |
| High-level Instruction | What motion should the car currently take to accomplish the instruction "Continue in a straight line along your current path until you reach the upcoming intersection."? |
| High-level Planner | Slightly below target speed, gently increase acceleration. Make a slight left turn. |
| Low-level Controller | Waypoint: [-0.1512451171875, -2.8828125], [-0.439697265625, -5.984375], [-0.71630859375, -9.1796875], [-1.0048828125, -12.4296875], [-1.201171875, -15.828125]

Visualization:
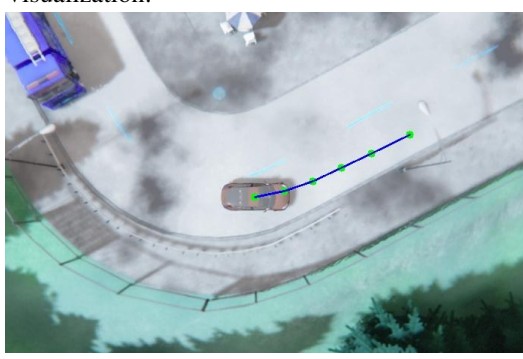 |

ment in transportation sectors and exacerbate privacy concerns related to the collection and utilization of vast amounts of personal data. Additionally, the deployment of such sophisticated systems could widen the digital divide, as access to and understanding of these technologies may not be equitable across all socioeconomic groups. It's crucial to address these challenges through thoughtful regulation, education, and inclusive design practices to ensure that the benefits of autonomous driving technologies are equitably distributed across society.

Table 5: AD-H performs well in complex nighttime turning environments, whereas LMDrive may result in the vehicle stopping in the middle of the road. The green dots in the figure represent waypoints. When a waypoint coincides with the vehicle's position, it indicates that the vehicle has come to a stop. Navigation Instruction: Upon covering [x] meters, a right turn at the traffic signal is mandatory.

| Time | AD-H | | LMDrive |
| --- | --- | --- | --- |
| | Driving command | Veritcal View | Vertical View |
| $T_0$ | Watch out for the car ahead, there's a vehicle in front. Apply brakes safely. |  |  |
| $T_1$ | Slow down to ensure safety. Make a slight right turn. |  |  |
| $T_2$ | Slightly below target speed, gently increase acceleration. Keep the steering wheel straight. |  |  |

Table 6: AD-H performs well in complex turning environments, whereas LMDrive may result in the vehicle stopping in the middle of the road. The green dots in the figure represent waypoints. When a waypoint coincides with the vehicle's position, it indicates that the vehicle has come to a stop. High-level instruction: Upon covering [x] meters, a right turn at the traffic signal is mandatory.

| Time | AD-H | | LMDrive |
|------|------|--|---------|
| | Driving command | Vertical View | Vertical View |
| $T_0$ | Approaching a junction, prepare to follow traffic rules. Slow down to ensure safety. Make a slight right turn. |  |  |
| $T_1$ | Approaching a junction, prepare to follow traffic rules. Slow down to ensure safety. Apply brakes safely. |  |  |
| $T_2$ | Approaching a junction, prepare to follow traffic rules. Slow down to ensure safety. Make a slight right turn. |  |  |

Table 7: Our method has stronger instruction following performance.

| High-level Instruction | Upon completing 10 meters, a left turn at the intersection is compulsory. | |
|------------------------|------|------|
| Method | LMDrive | AD-H |
| Vertical View |  |  |
| Mid-level Driving Command | None | Slow down to ensure safety. Make a slight left turn. |

Table 8: Full list of long-horizon instructions in LangAuto-Long-horizon benchmark.

| ID | Driving command |
|----|-----------------|
| 0 | Go straight ahead, turn left at the end of the road, then continue straight. |
| 10 | Go straight until the intersection ahead, then turn right, and continue along the road. |
| 12 | Go straight to the first intersection ahead and turn left, then continue straight. |
| 20 | Turn right ahead and then go straight. |
| 26 | Turn right ahead, go straight, then turn right again. |
| 34 | Go straight to the T-junction ahead, then turn left and follow the route. |
| 44 | Go straight to a crossroads, then turn left, then continue straight. |
| 46 | Go straight to the T-junction, turn right, and continue straight. |
| 48 | Follow the route, and continue straight when you reach the crossroads. |
| 57 | Go straight to the intersection where, on the left front side, there is an open space with some parked vehicles, and turn left. |
| 68 | Keep going along this road. |
| 70 | Turn left at the T-junction ahead, then follow the road. |
| 74 | Turn left ahead when you reach the cornfield, then turn left again when you encounter an open area. |
| 81 | Slightly turn left along the road ahead, then turn right, turn left at the T-junction, and then go straight. |
| 84 | Go straight until you see a turning point with palm trees ahead, then turn right and follow the road. |
| 88 | Turn right at the T-junction, go straight, then turn right at the T-junction where there are grid lines on the ground. Then continue straight. |

Table 9: Full list of the 26 different types of driving sub-commands in AD-H dataset. Combining sub-commands can result in over 170 variations of driving commands.

| Type | Driving command |
|------|-----------------|
| Perception | Approaching a junction, prepare to follow traffic rules. 
 A vehicle is present at the junction. Be cautious. 
 Multiple vehicles are present at the junction. Be cautious. 
 Watch out for the car ahead, there's a vehicle in front. 
 Watch out for the cars ahead, there are multiple vehicles in front. 
 A vehicle is present in the lane. Be cautious. 
 Multiple vehicles are present in the lane. Be cautious. 
 There is a bike ahead. Be cautious. 
 Multiple bikes are ahead. Be cautious. 
 There is a pedestrian ahead. Be cautious. 
 Multiple pedestrians are ahead. Be cautious. 
 There is a red light ahead. 
 There is a stop sign ahead. |
| Speed | Slow down to ensure safety. 
 Start accelerating gradually towards the target speed. 
 Remain stopped due to brake application. 
 Significantly below target speed, accelerate if safe. 
 Slightly below target speed, gently increase acceleration. 
 Above target speed, decelerate. 
 Maintain current speed to match the target speed. |
| Steer | Steer right sharply. 
 Make a slight right turn. 
 Steer left sharply. 
 Make a slight left turn. 
 Keep the steering wheel straight. |
| Break | Apply brakes safely. |

