# OpenReview forum: "AD-H: Autonomous Driving with Hierarchical Agents"
_ICLR.cc/2025/Conference — ICLR 2025 Conference Withdrawn Submission_

### Official Review · Reviewer_uYDL · 2024-10-31

**Soundness:** 3
**Presentation:** 3
**Contribution:** 2
**Rating:** 5
**Confidence:** 3

**Summary:**

This paper proposes AD-H, a hierarchical multi-agent system for autonomous driving that decomposes the driving task into two levels handled by separate agents: (1) a high-level MLLM-based planner that translates natural language driving instructions into mid-level driving commands, and (2) a lightweight controller that converts these commands into low-level vehicle control signals. The key insight is separating the high-level reasoning from low-level control, allowing the MLLM to focus on its strengths in language understanding and planning rather than directly generating control signals. The authors also contribute a new dataset called LMDrive-H with 1.7M frames annotated with hierarchical actions at multiple levels. Through extensive experiments in the CARLA simulator, AD-H demonstrates superior performance compared to state-of-the-art methods, particularly in generalizing to novel scenarios and long-horizon instructions.

**Strengths:**

- The hierarchical decomposition of driving tasks into high-level planning and low-level control is well-motivated and effectively leverages the strengths of MLLMs
- The mid-level driving commands provide an elegant interface between language understanding and vehicle control
- Clear and well-organized presentation
- Creation of a new dataset with hierarchical annotations

**Weaknesses:**

### Limited baseline comparisons：
- Should include more recent works like CarLLaVA and other hierarchical approaches
### Insufficient ablation studies:
- No analysis of impact of different granularities of mid-level commands

### Limited novelty
 The core idea of hierarchical decomposition is relatively straightforward and similar approaches have been explored in diverse domains, including autonomous driving and robotics control.

### The experimental validation could be more comprehensive:
- Only one main baseline (LMDrive)
- Limited analysis of failure cases


### Limited Real-World Validation:

- All experiments are conducted in simulation (CARLA)
- No discussion of potential sim-to-real transfer challenges


### Computational Requirements:


- The system requires two large models (7B parameter MLLM + 350M parameter controller). Runtime analysis and latency considerations are not discussed


### Dataset Construction:
- The rule-based annotation process for mid-level commands could introduce biases
- More details needed on annotation quality control measures

**Questions:**

1. Why not compare with other recent approaches like CarLLaVA or other hierarchical systems?
2. Have you analyzed how different levels of granularity in mid-level commands affect performance?
3. What is the computational overhead of the hierarchical system compared to end-to-end approaches?
4. How does the system handle conflicts between high-level instructions and mid-level commands?

---

### Official Review · Reviewer_bf8a · 2024-11-02

**Soundness:** 2
**Presentation:** 3
**Contribution:** 1
**Rating:** 3
**Confidence:** 5

**Summary:**

This paper introduces AD-H, a hierarchical framework for autonomous driving using multimodal large language models (MLLMs). Unlike conventional methods that translate high-level instructions directly into control signals, AD-H separates high-level planning from low-level control. The MLLM planner generates mid-level commands, which the controller then executes. This paper also provides an autonomous driving dataset with multi-level instructions and driving commands. Closed-loop evaluations show that AD-H outperforms existing methods in driving performance, exhibits self-correction capabilities, and demonstrates excellent generalization under long-horizon instructions and new environments.

**Strengths:**

1. The paper is clearly presented, easy to understand.
2. AD-H fully leverages the MLLM’s capabilities in perception, reasoning, and planning by focusing on generating mid-level commands rather than direct control signals, which maximizes the utility of the MLLM in autonomous driving.
3. The proposed system demonstrates impressive performance, including self-correction during operation, and outperforms current state-of-the-art methods in generalization, especially in new or complex driving scenarios.

**Weaknesses:**

1. The innovation of hierarchical planning is limited: DriveMLM has investigated the LLM-based mid-level lateral and longitudinal decision commands by setting decision states in Carla.
2. This paper uses LLaVA-7B and OPT-350M as the high-level planner and low-level controller, respectively. Can the 7B planner provide accurate mid-level instructions? Can the 350M controller output a well-planned trajectory based on ground truth instructions?
3. Separate evaluation experiments for each module should be added to better assess the performance of each module and the effectiveness of mid-level instructions in enhancing the controller.
4. Figure 3 demonstrates the self-correction capability of the MLLM controller, but what is the source of this capability? More examples should be provided to analyze the self-correction ability. Additionally, when training the MLLM controller, we aim for strong instruction-following capability. Can the current model balance instruction-following with self-correction effectively?
5. In Table 5, I understand that the ADH dataset used to train LM-Drive does not include mid-level instructions and only uses the resample method. The superior performance of ADH compared to LM-Drive may stem from the mid-level instructions in the dataset. An experiment should be added where mid-level instructions are input along with instructions to train LM-Drive for comparison, analyzing the advantages of using a low-level controller versus having LLaVA-7B generate trajectories directly.
6. Regarding model-related questions: After training the MLLM planner and controller separately, was joint training conducted? Are the decision frequency of the planner and the planning frequency of the controller consistent?
7. The hyperlinks for tables and figures are set incorrectly, such as in Tables 5 and 6, and Figures 4, 5, 6, and 7. There is also a hyperlink error in lines 312–313.

**Questions:**

Please see the weaknesses.

---

### Official Review · Reviewer_5tUR · 2024-11-02

**Soundness:** 3
**Presentation:** 3
**Contribution:** 3
**Rating:** 6
**Confidence:** 5

**Summary:**

This paper presents a hierarchical framework, AD-H, that leverages multimodal large language models (MLLMs) for autonomous driving in complex, dynamic environments. The framework introduces a dual-agent approach, with an MLLM planner responsible for perceiving environmental information and high-level instructions to generate mid-level commands, and a controller that executes these commands. This separation enables the MLLM planner to focus on high-level reasoning and planning, thus avoiding the limitations of directly mapping language to low-level control signals. Additionally, the authors contribute a new autonomous driving dataset annotated with hierarchical action levels, which supports the training of the AD-H system. Experimental results demonstrate AD-H’s stronger performance compared to state-of-the-art driving baseline (LMDrive), and robust generalization under long-horizon instructions and novel environmental conditions.

**Strengths:**

1. **Conceptual Simplicity and Design Insight**: The hierarchical agent framework, separating the MLLM planner and the controller, is a conceptually straightforward and interesting approach. The design highlights MLLMs’ strengths in high-level reasoning while minimizing their drawbacks in directly outputting low-level action.
2. **Dataset Contribution**: The new dataset with hierarchical action annotations is a valuable resource for advancing autonomous driving research, especially in tasks requiring a separation between high-level instructions and mid-level driving commands.
3. **Comprehensive Experiments**: The closed-loop evaluations cover a wide range of scenarios, highlighting AD-H’s strengths in generalization, and performance under long-horizon instructions and unseen conditions. The results showcase AD-H’s good performance over SOTA methods and emphasize the effectiveness of hierarchical planning.
4. **Clear and Engaging Presentation**: The writing and presentation are well-structured, with clear explanations of the hierarchical design and insights into MLLM limitations. The quantitative results and qualitative illustrations provide a convincing demonstration of the framework’s effectiveness.

**Weaknesses:**

1. **Evaluation of Consistency between Language and Action**: While the hierarchical framework is well-conceived, further evaluation of how consistent the language command and executed actions are, would be beneficial. In other words, do the final control actions comply with the mid-level commands? This is essential for fully validating the model’s real-world safety and applicability.

2. **Labeling of Mid-Level Instructions**: While the authors clearly introduced the hierarchical design, the methodology for labelling mid-level instructions lacks sufficient detail. For example, how do the authors define the high-level and mid-level instructions, and what is the principal difference between them? What is the design principle for mid-level instructions? While code snippets for generating mid-level commands are attached in the supplementary, descriptions of these design principles would better help readers grasp the gist of the design.

3. **Inclusion of Point Cloud in the Controller Only**: As shown in Fig 2, the choice to include point cloud data only within the MLLM controller, but not in the MLLM planner, is interesting but unexplained. More context regarding this decision would clarify whether it impacts the controller’s performance or if it’s just to balance computational resources.

4. **Ablation Study on the Motion Resampling**: In Section 6.1 and Table 5, the ablation study findings appear counterintuitive, as resampling is expected to improve performance but seems to lower certain metrics in LMDrive. An explanation regarding the discrepancy, and an additional experiment with AD-H trained without resampling, would possibly strengthen the study. It's fine that the resampling increases the performance, since that is also part of the contribution in this paper. And clarifying the performance contributions of each component would improve the message to the community regarding AD-H’s core strengths.

5. **Minor Presentation Issues**: A few minor presentation improvements are recommended, including correcting the typo of reference failure on Line 313 and marking the second-highest metrics in tables with an underscore to improve clarity.

**Questions:**

See the weakness section.

---

### Official Review · Reviewer_rPjF · 2024-11-04

**Soundness:** 2
**Presentation:** 3
**Contribution:** 1
**Rating:** 1
**Confidence:** 5

**Summary:**

The paper addresses the limitations of existing works in autonomous driving that implement large language models (LLMs) for directly translating high-level instruction into low-level control signals. The authors suggest that such a translation procedure, on the one hand, deviates from the original problem paradigm of language generation. On the other hand, it fails to leverage some of the promising features of the LLMs. Therefore, the authors proposed a hierarchical structure where LLMs act only as high-level perception and planning modules, generating natural language commands for controllers to execute. To make their work self-consistent, the authors built a driving dataset for evaluation.

**Strengths:**

1. From their statements, we can agree that the authors have made tremendous efforts to build a large autonomous driving dataset with multi-level driving command annotations to evaluate the model.
2. The ablation study of the experiment has been carefully designed to investigate influences from the training datasets and the different downstream controllers.

**Weaknesses:**

In short, the paper, overall, makes the most minuscule contribution to society. Here are some key problems in the fundamental logic and their experiment design:
1. The authors suggest that by decoupling the perception-planning and controlling stage, one can better leverage LLMs' *emergent capabilities*. However, their justifications are rather feeble - the hierarchy helps free the LLM from generating low-level control signal decoding and focus on perceptive reasoning and planning. From a methodology perspective, the decoupling of planning and control can introduce additional bias during message passing from upstream to downstream. For example, if the LLM upstream yields an arbitrary command such as "slow down to ensure safety," how can the authors make sure that the target speed generated by the downstream controller can guarantee safety? Meanwhile, during fine-tuning, how can you determine whether the controller should be further fine-tuned to improve control quality or the upstream LLM should be fine-tuned to improve command generation?
2. On the other hand, the authors attribute the limitations of previous methods to a cause that **they deviate from the inherent language generation paradigm**. However, existing works have proposed several methods as well as benchmark datasets to exploit and evaluate the reasoning capabilities of the LLMs in autonomous driving systems. For example, in the paper "LaMPilot: An Open Benchmark Dataset for Autonomous Driving with Language Model Programs" from CVPR 2024, the authors propose and provide a collection of atomic command building blocks with high-level objectives and use LLM to fill in blanks in these commands to achieve control. The authors should investigate and compare their work to this paper and other works that similarly showcase their method's superiority.
3. This paper's experiment design is rather poor. On the one hand, the authors built a dataset for self-consistently evaluating their proposed method but **only** compared their results to one of the existing methods. This is far from sufficient to justify their claimed benefits. Meanwhile, it is hard to convince the audience that the dataset is not built with a bias in favor of the proposed method over the competitors. Lastly, the primary concern from decoupling planning and controlling arises from extra time and memory requirements, which the authors didn't investigate further in their experiments.

**Questions:**

From the comments above, here are the main problems I have for the authors:
1. What theory supports your assertion that incorporating a hierarchical model that decouples perception, planning, and control can help a system built on LLM to achieve better reasoning and generalizability?
2. For your empirical experiment, can you compare other methods for training and evaluating your proposed dataset?
3. What are your method's average extra time and memory requirements compared to the counterpart that does not decouple the planner and controller? Is this trade-off truly worth it in real-life deployment?

---

### Official Review · Reviewer_Sw7q · 2024-11-10

**Soundness:** 3
**Presentation:** 3
**Contribution:** 2
**Rating:** 5
**Confidence:** 4

**Summary:**

This paper introduces a new paradigm of applying Multi Modality Large Language Model (MLLM) in autonomous driving motion planning. Compared to existing MLLM planning methods that tries to let the MLLM produce a plan directly from sensor inputs, this work divides the process into a hierarchy where a larger MLLM first produce a high level plan, followed by a smaller MLLM to produce a fine grained waypoint  embedding based on that high level plan. The author trains each part separately based on LMDrive dataset, with the larger model against driving intentions expressed in natural language and the smaller model against waypoints. The authors perform experiments against the baseline method LMDrive that directly produce waypoints from sensor inputs and observed improvements in driving quality.

**Strengths:**

There are existing works that applies MLLMs to autonomous driving and the hierarchical MLLM agent paradigm introduced by this paper is similar to the chain-of-thoughts technique in the LLM area. This paper is original in that it brings this chain-of-thoughts technique to MLLM based autonomous driving and shows advantage in generalization ability and driving behavior.

Overall the paper is easy to follow and clear written.

**Weaknesses:**

- The author did not demonstrate failure cases. On the LangAuto benchmark, the best route complete rate is 53.2%, what are the cases where it did not complete?

**Questions:**

- Please fix the `??` in line 312.
- Does the model intake video sequence of just single frame multi camera images?
- Does the model consume vehicle state information (such as historical speed)?

---

### Note · Authors · 2024-11-15

I have read and agree with the venue's withdrawal policy on behalf of myself and my co-authors.